# Sympathetic Modulation in Cardiac Arrhythmias: Where We Stand and Where We Go

**DOI:** 10.3390/jpm13050786

**Published:** 2023-05-01

**Authors:** Wei-Hsin Chung, Yen-Nien Lin, Mei-Yao Wu, Kuan-Cheng Chang

**Affiliations:** 1Division of Cardiovascular Medicine, Department of Medicine, China Medical University Hospital, Taichung 40447, Taiwan; 2UCLA Cardiac Arrhythmia Center, Ronald Reagan UCLA Medical Center, Los Angeles, CA 90024, USA; 3School of Medicine, China Medical University, Taichung 404333, Taiwan; 4School of Post-Baccalaureate Chinese Medicine, China Medical University, Taichung 404333, Taiwan; 5Department of Chinese Medicine, China Medical University Hospital, Taichung 40447, Taiwan

**Keywords:** sympathetic modulation, arrhythmias

## Abstract

The nuance of autonomic cardiac control has been studied for more than 400 years, yet little is understood. This review aimed to provide a comprehensive overview of the current understanding, clinical implications, and ongoing studies of cardiac sympathetic modulation and its anti-ventricular arrhythmias’ therapeutic potential. Molecular-level studies and clinical studies were reviewed to elucidate the gaps in knowledge and the possible future directions for these strategies to be translated into the clinical setting. Imbalanced sympathoexcitation and parasympathetic withdrawal destabilize cardiac electrophysiology and confer the development of ventricular arrhythmias. Therefore, the current strategy for rebalancing the autonomic system includes attenuating sympathoexcitation and increasing vagal tone. Multilevel targets of the cardiac neuraxis exist, and some have emerged as promising antiarrhythmic strategies. These interventions include pharmacological blockade, permanent cardiac sympathetic denervation, temporal cardiac sympathetic denervation, etc. The gold standard approach, however, has not been known. Although neuromodulatory strategies have been shown to be highly effective in several acute animal studies with very promising results, the individual and interspecies variation between human autonomic systems limits the progress in this young field. There is, however, still much room to refine the current neuromodulation therapy to meet the unmet need for life-threatening ventricular arrhythmias.

## 1. Introduction

The link between the brain and the heart was initially mentioned in 1628 by William Harvey “For every affection of the mind that is attended with either pain or pleasure, hope or fear, is the cause of an agitation whose influence extends to the heart.” [1]. In the 17th century, Thomas Willis first described the vagi and its cardiac projections. This work was followed by Eduard and Heinrich Weber in the 18th century, when they pointed out that right vagal stimulation slows the heart rate. The interaction between the central nervous system and the cardiac nervous system (CNS) has been studied and found to be increasingly complex over the past 400 years. Autonomic dysregulation enhances the risk of ventricular arrhythmias (VAs) and sudden cardiac death (SCD) and is commonly known as sympathetic overactivity and parasympathetic withdrawal [2]. The interest in neurocardiology has merged to become an attractive therapy for the unmet need for antiarrhythmic therapy for atrial arrhythmias, VAs, and heart failure [3,4]. VAs are life-threatening ventricular electrical abnormal activities that predispose to a major cause of SCD. Cardiac neural control was modulated by parasympathetic output from the vagal nerve and sympathetic output via the intrathoracic ganglia. Sympathoexcitation induces changes in ventricular repolarization and reduces VA thresholds [5]. In explanted hearts from patients with prior VAs, enhanced sympathetic nerve sprouting was found in the border zone of the scar and survived the myocardium [6]. Importantly, sympathetic hyperinnervation alone without coronary artery disease can increase susceptibility to VAs [7]. In this review, we will focus on how sympathetic activities influence cardiac electrophysiology, currently known clinical studies, and the future direction of sympathetic modulation.

## 2. Neuroanatomy of the Sympathetic System

The CNS comprises central components (Level 3), intrathoracic extracardiac components (Level 2), and the intrinsic cardiac nervous system (ICNS) (Level 1) [8]. Level 1 represents the “little brain on the heart” system. The ganglionated plexi resides near the junction between the fat pad and myocardium and comprises the afferent, efferent, and local circuit neurons. Level 2 is the intrathoracic extracardiac components, which are the cervicothoracic ganglia, which connect the central components and the ICNS and include the parasympathetic and sympathetic systems. Level 3 is the extrathoracic and extracardiac components, including the dorsal root ganglia, nodose ganglia, and areas in the spinal cord and brainstem controlling the heart. Functionally, neurons can be classified into afferent (sensory), efferent (motor), or local circuit neurons [9].

The preganglionic neurons of the sympathetic component reside in the intermediolateral columns of the spinal cord and project via C7–T6 to the superior cervical, middle cervical, stellate, and cervicothoracic ganglia and synapse with postganglionic nerves [10]. Postganglionic neurons then project from the ganglia to the atrium and ventricles along the epicardial vessels and reach the endocardium. The fibers innervate the atria more than the ventricles and the base more than the apex [11]. The preganglionic neurons release acetylcholine to bind the nicotinic acetylcholine receptors, which then activate postganglionic neurons, which release norepinephrine. Notably, epinephrine is rarely released [12]. The complex yet elegant autonomic system network modulates the cardiac physiological and electrophysiological functions from beat to beat and second to second.

## 3. Assessing Autonomic Nervous System (ANS) Function 

With the growing interest in autonomic modulation therapy, the assessment of ANS function has been receiving attention. 

### 3.1. Heart Rate Variability (HRV)

HRV indicates the beat-to-beat oscillation in the R-R interval. When analyzed by the power spectral density of the R-R interval variability, two major components in power can be obtained [13]. The concepts of low-frequency (LF, 0.04–0.15 Hz) and high-frequency (HF, 0.15–0.4 Hz) to describe low and high heart rate oscillation, respectively, were then developed in the 1980s [14]. It was postulated that LF and HF correspond to sympathetic and parasympathetic functions, respectively. However, more than 29,000 HRV-related studies have been published and show the limitations of HRV as an index of ANS function [15]. The long-term, 24 h LF/HF ratio is influenced by the lying ratio during the 24 h analysis and cannot be interpreted in the same way as the short-term, 5 min analysis. The cutoff of 0.15 Hz was proposed by Sayers and based on dog experiments [16,17]. There is no convincing evidence showing this value applies to humans under all conditions. Most importantly, the LF component, previously used as an indicator of sympathetic functions, has now been denied by numerous studies [15] and has been shown not to relate to sympathetic innervation quantified by positron emission tomographic imaging [18]. 

### 3.2. Baroreflex Sensitivity

BRS is a physiological reflex to prevent short-term, wide fluctuations of arterial blood pressure. It is therefore measured by the reflex changes in the R-R interval in response to induced changes in blood pressure in a pharmacological (phenylephrine) challenge, the Valsalva maneuver, or mechanically manipulating the neck chamber. Early animal experiments have found that the reduced BRS is related to ventricular fibrillation [19]. This observation was also shown in the ATRAMI study, in which a depressed BRS (<3.0 ms/mmHg) was found to be an independent predictor of cardiac mortality after myocardial infarction. The evidence also exists in HF, where a depressed BRS was found to be an independent predictor of cardiac death [20]. Importantly, spontaneous BRS can be obtained non-invasively and has been investigated recently to show the prognostic value of cardiac death in stable heart failure patients [21]. This finding is of great importance in opening a new way to address the impact of BRS. 

### 3.3. Heart Rate Turbulence (HRT)

HRT was initially introduced in 1999 by Schmidt et al. [22]. In normal settings, after the compensatory pause caused by a premature ventricular complex, the sinus rate accelerates and slows thereafter [22]. It was found in the ATRAMI study that the onset and slope of HRT, when combined with BRS and HRV, can be stronger predictors of cardiac arrest and nonfatal cardiac arrest [23]. 

### 3.4. T-Wave Alternans (TWA)

TWA was initially reported in 1994 by Rosenbaum et al. [24]. They found that in patients with low-level electrical alternans (beat-to-beat change in amplitude < 15 microV) after myocardial infarction, the mortality and VAs risk were higher. Furthermore, in patients with preserved ejection fraction after myocardial infarction, the prevalence of TWA was low but associated with VA events [25]. TWA reflects the heterogeneity or dispersion in ventricular repolarization. The absence of TWA predicts freedom from VT and VF. 

## 4. Situational Conditions Related to Sympathoexcitation and Ventricular Arrhythmias

Emotional stress, especially anger and fear, is known to play a significant role in fatal arrhythmia induction and termination, as evidenced by animal models and human studies [26,27]. Existing studies have shown the impact of emotion on the ECG [28]. Specifically, in patients with implantable cardioverter-defibrillators, T-wave alternans could be induced by anger triggered in a laboratory setting and were found to predict future VAs [27,29]. Although poorly understood, there was emerging evidence showing that the interplay between the central nervous system and the cardiovascular system might have an important role [30]. The proposed mechanisms include alternation in the balance of sympathetic and parasympathetic [31], the spatial distribution of cardiac autonomic input [32], and the surge of plasma catecholamines. 

Exercise training is considered an effective non-pharmacological treatment for several cardiovascular diseases [33], and the benefits have been shown in myocardial infarction patients and animals [34]. The mechanism, intensity, and modality remain to be elucidated. It is hypothesized that exercise could rebalance the sympathetic and parasympathetic systems by restoring LF/HF [35]. Molecularly, exercise inhibits CaMKII-dependent RyR2 hyperphosphorylation to reduce the calcium leak during diastole, with a result of decreased afterdepolarization and suppressed VAs [35,36]. On the other hand, contradictory evidence increasingly reported that high-intensity chronic exercise might, in fact, be harmful, with evidence of the increased sudden cardiac death incidence in a certain modality of sport, i.e., basketball players [37]. The possible underlying mechanisms include a prolonged QTc interval and actional potential duration resulting from a reduction in I_to_ current, increased ventricular fibrosis, and enhanced HCN4 expression [38]. In totality, future studies are warranted to provide a beneficial and individualized exercise program. 

## 5. Sympathoexcitation and Regional Electrical Property Modulation

Sympathoexcitation is known to mediate inotropy, chronotropy, dromotropy, and lusitropy. In the presence of diseased or injured myocardium, the autonomic system could adversely impact electrical stability [39]. Sympathoexcitation increases the VA’s susceptibility by shortening the action potential via slowing the inward rectifying potassium channel (IK_S_), increasing repolarization heterogeneity, intracellular calcium, and after depolarization by augmenting calcium influx to the sarcoplasmic reticulum [40]. The increased intracellular calcium thereby results in early afterdepolarization via reactivating L-type calcium channels. The augmented calcium influx predisposes to spontaneous calcium release from the sarcomere, followed by sodium-calcium exchange, and induces an early or delayed afterdepolarization [41]. In addition to neuronal stimulation, sympathoexcitation also activates alpha2 adrenergic receptors and increases gap junction expression [42]. Although earlier studies concluded that sympathetic stimulation did not influence ventricular activation patterns [43]. However, despite the misunderstanding that sympathetic nerves innervate both ventricles equally, it has been clearly shown by Vaseghi et al. that LSGS increases regional repolarization dispersion in the LV anterior wall and apex, whereas RSGS influences mainly the RV posterior wall [44]. 

## 6. Sympathetic Remodeling after Myocardial Infarction 

Cardiac insult, i.e., myocardial infarction, results in scar formation, breaks down cardiac autonomic regulation, and remodels the sympathetic innervation [45]. The change at the organ level alters the electrical propagation and serves as the substrate for reentrant arrhythmias. Systemically, the injury results in sympathoexcitation and vagal withdrawal, leading to afferent-mediated activation. The abnormal cardiac afferent signaling continues and can lead to cardiac disease and fatal arrhythmias [46]. Infarction also causes sympathetic denervation within the scar, and this process begins within 20 min after coronary occlusion [47]. This results in an abnormal response to sympathoexcitation. More importantly, the surviving myocardium demonstrates heterogenous effective refractory period (ERP) shortening induced by stellate ganglion stimulation and norepinephrine infusion [48]. These areas also demonstrated the exaggerated ERP shortening induced by norepinephrine when compared with the normal myocardium basal to the infarct. Evidence from metaiodobenzylguanidine also shows nerve denervation and reinnervation in human studies in both ischemic and nonischemic cardiomyopathy [49,50].

Sympathetic nerve sprouting and hyperinnervation are arrhythmogenic. The axon regeneration was triggered by neuronal growth factor (NGF) produced by the surrounding myocardium and slowly reached a constant speed after three days [51]. The uncontrolled hyperinnervation, however, could be arrhythmogenic. Cao et al. had shown that in the explanted hearts, the sympathetic nerves were significantly increased peripherally to the necrotic tissue in patients with a history of VAs [6]. This was further supported by the study showing that in hypercholesterolemic rabbits, there are high densities of growth-associated protein 43 and tyrosine hydroxylase. The condition group also demonstrated longer QTc intervals, more QTc dispersions, and higher episodes of ventricular fibrillation [7]. In dogs without ischemic heart diseases, rapid pacing induces hyperinnervation and a higher risk of sudden cardiac death [52].

The remodeling also develops at the neurochemical level. Adrenergic neurons transdifferentiating to cholinergic neurons was reported in the hypertensive heart failure rat model and the low-level vagal stimulation model [53,54]. This was further supported by the evidence that the chronic ischemic porcine model demonstrated a cholinergic-to-adrenergic trans-differentiation in bilateral stellate ganglions [55]. The finding indicates that the injury is “sensed” equally in bilateral neuroaxis and highlights the importance of bilateral neuromodulation.

## 7. Clinical Implications

The autonomic imbalance interacts with the electrical instability of the myocardium, and therefore, neuromodulatory therapies aim to restore this balance. Currently, known interventions can be designed to decrease sympathetic efferent signaling and/or increase parasympathetic activity. This review will focus on the sympathetic modulation in cardiac arrhythmias. Table 1 is the summary of the current animal and clinical studies regarding the clinical implications of neural modulation.

### 7.1. Transepidural Anesthesia (TEA)

Although bilateral stellate ganglion block (SGB) is getting popular, TEA has the potential to provide a more complete sympathetic blockade. By injecting anesthetics in the high thoracic epidural space (C8–T4), TEA blocks the fibers derived from T1–T4, which form the cardiac accelerator fibers, and C8, which form part of the inferior cardiac sympathetic nerve. Thereby, TEA reduces the incidence of VAs acutely by lengthening ventricular electrical repolarization and prolonging the ventricular effective refractory period [56,57].

However, this procedure still has limitations. Firstly, there are no reliable parameters to indicate an effective TEA. Secondly, high thoracic anesthesia could theoretically diminish cardiac contractility. However, Wink et al. demonstrated that TEA did not significantly diminish exercise-induced increases in cardiac function despite decreased baseline right and left ventricular systolic function [58]. One of the largest clinical case series also reported minimal effects of TEA on hemodynamic function [59]. Therefore, additional studies of TEA are warranted to investigate this undervalued technique.

### 7.2. Renal Nerve Denervation (RND)

RND was initially introduced as a treatment for resistant hypertension, and its effect was examined in randomized clinical trials [60]. The procedure was performed by radiofrequency ablation of bilateral renal afferent and effect nerves with a spiral catheter. Recently, the implication has been expanded to the treatment of atrial fibrillation with hypertension [61] and refractory VAs. The implication of RND on VAs was first described by Ukena et al. in two patients with refractory VT [62] and subsequently by Evranos et al. in 16 patients who underwent catheter ablation [63]. Recently, Bradfield et al. reported the clinical benefit of RND in patients with refractory VAs after CSD, which comprises mostly NICM patients [64]. The effect of RND is based on its therapeutic potential to decrease circulating catecholamines [65] and the supersensitivity to catecholamines in the scar border zone. Other studies report stellate ganglion remodeling after RND [66]. 

There are several limitations to RND. Firstly, the distribution of renal nerves is heterogenous, with fewer nerves over dorsal and distal renal arteries [67]. Secondly, mounting evidence has shown that sympathetic denervation results in lower neuropeptide Y and tyrosine hydroxylase by 4 days after denervation, but after 6 weeks to 6 months, reinnervation is observed in both large and small animal models [68,69]. The phenomenon of reinnervation is not surprising given the fact that the soma was not targeted in the procedure. Lastly, there is no endpoint to the standard RND procedure [70,71].

### 7.3. Cardiac Sympathetic Denervation (CSD) [72]

CSD can be performed as a minimally invasive procedure or as a video-assisted surgery. CSD disrupted both afferent and efferent fibers [73,74]. Electrophysiologically, bilateral CSD mitigates the repolarization heterogeneity in terms of Tp-Tend and T-wave dispersion in infarcted porcine models. Prior studies had shown left-side CSD to be beneficial in long QT syndrome and catecholaminergic polymorphic VT in multicenter studies [75,76]. A single-center study evaluating 41 patients with a VT storm showed a >80% reduction in ICD shock and 48% ICD shock-free survival in one year. However, the survival rate is lower in patients with only left-side CSD [77]. Later, a multicenter study also confirmed the effect of CSD in ICM and NICM. The investigators also found that left-sided CSD, longer VT cycle lengths, and worsening heart failure status (NYHA III or IV) are important predicting factors of VT recurrence. Although the ideal timing of CSD is unknown, the relationship between advanced heart failure status and the poor outcome of catheter ablation has been confirmed in previous studies [78]. These findings highlight the importance of an earlier approach to preventing the vicious cycle of heart failure and VAs. The complications reported in a systemic review of 141 patients were hypotension (9%), pneumothorax (5%), Horner’s syndrome (3%), and sweat pattern change (3%) [79]. Horner’s syndrome resolved completely in four of five patients within 6 months. Hypotension developed in the postoperative period, and all patients have been weaned off vasopressors. However, further studies are required to elucidate the temporal changes of heart failure, stellate ganglion remodeling, and the timing of intervention.

### 7.4. Stellate Ganglion Block (SGB)

While CSD offers a permanent treatment, it is an invasive procedure. In contrast, SBG was initially developed for managing complex regional pain syndrome [80] but has shown promising results in treating VAs. SGB can be performed at the bedside without sedation or interruption of anticoagulation, guided by ultrasonography, to suppress sympathetic activity temporarily through local anesthetic agents. Retrospective studies with a small number of patients have demonstrated the safety and efficacy of single-injection SGB in reducing VAs within 48 h [81,82]. Furthermore, the effect is consistent regardless of the type of cardiomyopathy or VA morphology. Similar to TEA, there is currently no clear endpoint for a successful SGB. Commonly used parameters include the temperature rise in the ipsilateral arm, Horner’s syndrome, and the perfusion index [83]. Possible complications include neck hematoma, transient hoarseness (caused by blocking the recurrent laryngeal nerve), or local anesthetic systemic toxicity, although the case series (n = 20) from Duke University demonstrated the safety of bilateral SGB. Recently, the continuous infusion approach of SGB was shown to be more effective than a single injection (n = 9 vs. 9) [84]. Prospective randomized trials are required to optimize patient selection, the timing of intervention, and procedure endpoints.

### 7.5. Carotid Baroreceptor Stimulation

The carotid sinus is an area located at the beginning of the internal carotid artery, containing baroreceptors and functioning as a modulator for maintaining blood pressure. Afferent nerves of baroreceptors originate from glossopharyngeal nerves and are located in the thickened wall of the carotid sinus and aortic arch. By activating the afferents, parasympathetic efferents are stimulated and sympathetic afferents are inhibited. Although studies have demonstrated increased susceptibility to ventricular arrhythmias in conscious dogs with healed MI and decreased baroreflex sensitivity [85], Liao et al. demonstrated that low-level electrical carotid baroreceptor stimulation with 80% of the threshold needed to slow the heart rate has an antiarrhythmic effect on ventricular arrhythmias by stabilizing ventricular electrophysiological properties in dogs [86]. Recent implications of baroreceptor stimulation were in HF. The randomized control trial BeAT-HF has shown the benefit of the BAROSTIM NEO system (CVRx, Minneapolis, Minnesota) in improving quality of life, exercise capacity, and N-terminal pro-B-type natriuretic peptide in HF with a reduced ejection fraction [87]. However, no clinical studies so far have shown antiarrhythmic efficacy in humans. 

### 7.6. Subcutaneous Nerve Stimulation (ScNS)

This novel technique was proposed by Chen et al. and is based on the neuroanatomy of the stellate ganglion, which innervates the heart and the skin of the neck and upper thorax. In the canine models, the implanted vagal nerve stimulating devices stimulate the subcutaneous sympathetic nerves over the Xinshu acupoint and the left lateral thoracic nerve for 2 weeks. They reported significant stellated ganglion remodeling with atrial tachycardia suppression [88]. The study was followed by ScNS delivered with blindly inserted electrodes in canine models, with a result of reduced atrial fibrosis in persistent atrial fibrillation [89]. Currently, they are conducting a prospective randomized clinical trial (NCT04529941) to determine the effect. This novel method, however, awaits further examination.

## 8. Future Directions

### 8.1. Pharmacological Sympathetic Modulation

Beta-blockers have been the earliest-used pharmacological sympathetic modulators since the 1950s. It is also the only drug demonstrating survival benefits in SCD. Beta-blockers are known to be classified as nonselective (i.e., propanolol) or selective (i.e., bisoprolol). However, it should be noted that beta-blocking properties are not limited to beta-blockers. Class I and III antiarrhythmic drugs also share this property. Importantly, beta-blockers are insufficient to suppress VAs despite being at a maximal dose.

Ivabradine, an I*_f_* blocker, has been shown to improve clinical outcomes in settings of congestive heart failure and coronary artery disease [90] and has been shown to reduce sympathoexcitation with evidence of improving symptoms of postural orthostatic tachycardia syndrome and inappropriate sinus tachycardia [91,92]. By inhibiting hyperpolarization-activated cyclic nucleotide-gated channels, Ivabradine slows diastolic depolarization, resulting in heart rate slowing. When used in high doses, it can also inhibit I*_Kr_* and lead to the prolongation of actional potential duration, which might be arrhythmogenic in settings with long QT. Despite the existing concern, studies including BEAUTIFUL, SIGNIFY, and SHIFT did not show an arrhythmogenic effect [93]. In addition, some studies suggest that Ivabradine might modulate the vagal effect with the result of preventing atrial arrhythmias [94,95]. Its antiarrhythmic or arrhythmogenic effects await further studies.

Neuropeptide Y (NPY) has emerged as a promising target for next-generation pharmacological neuromodulation. This transmitter is released during high-level sympathetic stimulation. Its effects include parasympathetic inhibition [96], myocardial calcium loading increase [97], and vasoconstriction [98]. It has also been found that in patients with heart failure, NPY ≥ 130 pg/mL indicates a worse outcome [99]. Additionally, in patients who underwent percutaneous coronary intervention, a higher NPY level is associated with a higher rate of recurrent VAs [100]. The currently developing BIBO 3304, an NPY Y1-receptor antagonist, has shown efficacy in neutralizing pro-arrhythmic conductions and stabilizing cardiac electrophysiology [101]. This awaits future studies.

### 8.2. Aortorenal Ganglion (ARG) Ablation 

Anatomically, the left ARG is located inferiorly to the superior mesenteric artery and abuts the posterosuperior aspect of the left renal vein. The right ARG is located between the IVC and descending aorta, superior to the right renal artery. Both ARGs are located posterior to the IVC. 

Neuroanatomically, ARGs exchange fibers with the celiac ganglion, and fibers also transit through the ARG to the superior mesenteric ganglion, although primary inputs to the superior mesenteric ganglion originate lower (T12–L1) than ARG inputs [102].

The concept of targeting ARG was initially proposed by Qian et al. [71]. They demonstrated in sheep that transvascular ARG high-frequency stimulation (10 Hz, 25 mA) could induce a hypertensive response and ipsilateral renal artery vasoconstriction. Later, Hori et al. showed the effectiveness of targeting ARG compared with targeting renal nerves. [103]. They reported that the hemodynamics are substantially influenced by ARG stimulation, and by ablating the renal artery nerves, the ARG simulation effect cannot be eliminated. Most importantly, ARG ablation provides a cardioprotective effect against ischemic-induced VAs compared with conventional RND. Further studies are required to translate the procedure into humans.

### 8.3. Resiniferatoxin (RTX) Deafferentation 

Myocardial infarction activates the cardiac adrenergic reaction, which drives ventricular arrhythmogenesis. The afferent limb of the cardiac sympathetic afferent reflex (CSAR) is mediated partially by the transient receptor potential cation subfamily V member 1 (TRPV1) channel [104]. RTX is a potent, selective, and irreversible activator of the TRPV1 channel. By epicardially applying RTX in the rodent infarct model, the CSAR was abolished. This resulted in a lower left ventricular end-diastolic pressure and prevented the progression of heart failure after myocardial infarction [105]. Later, the translation value of this approach in VA management was investigated by Koji et al. [106]. Following myocardial infarction, RTX was applied epicardially. Without altering the infarct border zone electrophysiology and histology, RTX deafferentation attenuated VA inducibility and the progression of heart failure. This novel therapy required further studies to confirm the dose and time of intervention before translating this novel therapy into a clinical procedure.

### 8.4. Modified CSD

Despite the fact that CSD has been shown to be effective in controlling VAs, those with poor cardiac and pulmonary status might not be ideal candidates. Inability to tolerate single-lung ventilation, presence of pericardial adhesion, and systemic anticoagulation are some of the reasons [107]. Recently, Cauti et al. reported a modified approach by applying 80 w unipolar radiofrequency to the proximal sympathetic chain (T2–T5) in five patients, sparing the stellate ganglion. This modified approach shortened the procedure time down to 18 ± 11 min and resulted in a substantial reduction in VAs. Although two patients developed VT recurrences early after the procedure, they still did well in the long term. This finding is in line with a recent case series including 20 patients who underwent bilateral CSD with a follow-up of 4 years [108]. Whether persevering with the stellate ganglion in CSD provides the same benefit as standard CSD is unknown. 

Additionally, currently, existing stellate ganglion radiofrequency ablation techniques and stereotactic radiosurgery are performed by anesthesiologists for patients with acute digital ischemia, a complex regional pain syndrome, or refractory chest pain [109,110,111]. These techniques, however, have not been used to manage VAs. On the other hand, the first-in-human cryothermal left-side stellate ganglion ablation in a patient with an electrical storm has recently been reported [112]. These less invasive, modified SGB and CSD procedures will require more attention and scientific evaluation in the future. 

### 8.5. Axonal Modulation Therapy (AMT)

Although ICD remains the main therapy for patients with VAs, it treats symptoms rather than the root cause of the disease [92]. AMT is a developing therapy that aims to provide selective and on-demand control of cardiac sympathetic activity. The use of peripheral nerve electrical stimulation is increasing, but selective blockage of axonal action potentials was recently discovered [93]. Currently, there are two primary protocols for blocking action potential propagation: kilohertz frequency alternating current (KHFAC) and direct current. By delivering KHFAC, the axonal membranes are forced into a state that does not favor action potential propagation [113]. One of the proposed mechanisms involves increasing the inward sodium current compared to the outward potassium current which results in dynamic membrane depolarization [114]. Studies have shown that the effective blocking frequencies range from 200 Hz to 30 kHz [115] with a sinusoidal or square wave, and the effective blocking amplitude ranges from 0.3 to 10 mA [116]. In the preliminary studies by Chui et al., the author utilized a charge-balanced direct current (CBDC) to selectively block the T1–T2 region and thereby increase cardiac electrical stability and reduce VA susceptibility in a porcine model [117]. Furthermore, they also showed that the scalability, sustainability, and memory properties of AMT and this proof-of-concept study serve the future development of an on-demand sympathetic block device [118].

### 8.6. Magnetic and Ultrasound Stimulation

Transcranial magnetic stimulation has already been used for treating pain and depression [100]. It has also been found to impact cardiac rhythm [101] and heart rate variability [102], which are considered surrogates of the autonomic system. Electromagnetic stimulation of parasympathetic trunks was previously reported to suppress atrial fibrillation [103]. This was followed by Wang et al., who used low-frequency electromagnetic stimulation on the left stellate ganglia in canines with acute myocardial infarction and demonstrated the benefit of suppressing ventricular arrhythmias [104]. Similarly, in a case series of five patients with electrical storms who were all treated with transcutaneous magnetic stimulation of the left stellate ganglion, the burden of ventricular arrhythmias decreased without adverse events [105]. Lastly, the same group used low-level ultrasound stimulation of the left stellate ganglion for 10 min in dogs following a myocardial infarction and demonstrated reduced left stellate ganglion activity along with a decrease in myocardial infarction-induced ventricular arrhythmias [119]. 

### 8.7. Glial Cell Modulation 

Glia are non-neuronal cells that exist throughout the central and peripheral nervous systems. They have been receiving attention for their role in contributing to the pathophysiology of various conditions [107]. Satellite glial cells envelop neurons and influence synaptic activity [108]. Recent research has found that satellite glial cells are enlarged and upregulate the Gfap in patients with recurrent ventricular arrhythmia, suggesting therapeutic potential. A designer drug (DRE-ADD) has been shown to activate satellite glial cells, leading to an increase in heart rate and blood pressure [109]. These findings suggest that the activity of satellite glial cells could be targeted to modulate sympathetic output and serve as a therapeutic target [120].

**Table 1 jpm-13-00786-t001:** Summary of animal and clinical studies of neuromodulation intervention.

Interventions	Indication	Mechanisms	Current Implementation	Notes
**Pharmacological**				
Beta-blocker (BB)	VA, SCD, and heart failure	Inhibits sympathetic effects by blocking BB	Well established	Some patients have VAs despite maximal doses.
NPY-receptor blocker [101]	Inhibit sympathoexcitation	Inhibits sympathetic effects by blocking NPY receptors (BIBO3304)	Acute large animal study	- UUnknown antiarrhythmic property.
**Stellate ganglion modulation**				
CSD via endoscopy assistance [77]	Refractory VA in Long QT, CPVT, ICM, and NICM	Disrupting neural transmission through SG by removing the sympathetic chain and SG	Survival benefit in multileft retrospective studies	- IInvasive - BBilateral CSD is more effective.
**Modified CSD**				
Radiofrequency ablation of the sympathetic chain [107]	Refractory VA In mixed etiology	Disrupting sympathetic chain neural transmission by radiofrequency ablation	1 clinical case series	- LLess procedure time. - PReserving SG may increase recurrence. - NNo endpoints.
Cryothermal ablation [112]	ES	Disrupting sympathetic chain neural transmission by cryothermal damage to the cervical SG	1 case report	Has therapeutic potential.
Magnetic stimulation of SG [121,122]	ES	Long-term depression and long-term potentiation change synaptic plasticity	Case series of 5 Acute large animal model	No histology or electrophysiology data.
Ultrasound stimulation of SG [119]	VA	Unknown mechanism. Possibly through thermal effects and anti-inflammatory effects	Acute large animal model	Unknown mechanism.
Stellate ganglion block (SGB) [82]	ES, bailed-out therapy Pain	Pharmacological temporal inhibition of cervical stellate ganglia nerve transmission to the myocardium	- MMultiple clinical studies - A bridge to permanent therapy	- NNo procedure endpoints. - NNo standard dose - Bilateral should be more effective, but it raises the concern of respiratory distress.
Transepidural anesthesia [59]	ES, bailed-out therapy Pain	Pharmacological temporal inhibition of cardiac afferents and sympathetic efferents at levels C8–T4	- CClinical retrospective study - AAcute large animal study - A Bridge to permanent therapy	- NNo procedure endpoints. - TTheoretically more effective than SGB. - RRisk of intrathecal bleeding.
**Therapies beyond the cervicothoracic SG and sympathetic chain**				
Renal nerve denervation [64]	- RRefractory VA after CSD - HTN (controversial result) - AF + HTN	Disrupting sympathetic nerves to modulate the circulating catecholamine release	- CClinical retrospective studies - CChronic and acute animal studies	- NNo procedure endpoints. - PReserved ganglion might cause recurrence.
Aorticorenal ganglion (ARG) ablation [103]	- VA	Ablating the ganglion is more effective than ablating the renal nerves.	- AAcute large animal study	- UUnknown safety in ablating venous or artery structures in humans.
Low-level baroreceptor stimulation [86]	- VA (animal) - HF - HTN	By activating the afferents, the parasympathetic efferents are stimulated and the sympathetic efferents are inhibited.	- AAcute large animal study	- CClinical used in HTN and HF. - BBenefit in VA has not been reported.
Epicardial RTX deafferentation [106]	- VA	Selectively and irreversibly activates the TRPV1 channel and then blocks the cardiac afferent reflex.	- AAcute large animal study	- RTX causes adhesion. - HHigh risk of bleeding during myocardial infarction.
Axonal modulation therapy [118]	- VA	KHFAC blocks action potential propagation	- Acute large animal study - ((Proof-of-concept)	- DDevices remain in prototype status.
Glial cell modulation [120]	- SSympathoexcitation	A designer drug (DRE-ADD) can cause sympathoexcitation by activating SG glial cells	Acute small animal study (Proof-of-concept)	- EEmerging therapeutic target.

AF = atrial fibrillation; CSD = cardiac sympathetic; ES = electrical storm; HF = heart failure; HTN = hypertension; KHFAC = kilohertz frequency alternating current; RTX = resiniferatoxin; VA = ventricular arrhythmia; denervation; NA = not available; SCD = sudden cardiac death; SG = stellate ganglion; VA = ventricular arrhythmia. Central Illustration: Shown here are the efferent (red line) and afferent (blue line) courses of the cardiac sympathetic afferent reflex with a multilevel treatment strategy. The cell body of afferent fibers is located in the dorsal root ganglia of C8–T9, especially T2–T6, and converges the information to the brainstem, including the nucleus of the solitary tract, rostral ventrolateral medulla (RVLM), and paraventricular nucleus. Baroreflex stimulation therapy enhances the afferent pathway and inhibits the sympathetic outflow. The paraventricular nucleus projects to the RVLM with integrated information and then to the intermediolateral column of the spinal cord. The preganglionic neurons course to the heart through the ventral rami and synapse on postganglionic primarily within extracardiac intrathoracic ganglia, including the superior, middle cervical, and mediastinal ganglia. The cardiac sympathetic output can be blocked by electromagnetic stimulation, stellate ganglion block, stellate ganglion ablation, cardiac sympathetic denervation, and axonal modulation therapy. Preganglionic fiber derives from T10–T11 and synapses with postganglionic fiber at the aorticorenal ganglion. This pathway can be modulated by aorticorenal ganglion and renal nerve denervation to decrease catecholamine release. AMT = axonal modulation therapy; ARG = aorticorenal ganglion; BRS = baroreceptor stimulation; CSD = cardiac sympathetic denervation; EMS = electromagnetic stimulation; NPY = neuropeptide Y; SGB = stellate ganglion block; TEA = trans epidural anesthesia; NTS = nucleus tractus solitarius; RVLM = rostral ventrolateral medulla.

## 9. Conclusions

The elegance of the cardiac neuraxis and its interaction with cardiac physiological and electrical functions have been shown in several studies. The role of sympathetic control and ventricular arrhythmogenesis is increasingly elucidated. These findings have been translated into targeted therapies. Currently, stellate ganglion block and cardiac sympathetic denervation are the most commonly used, and as a last resort, neuromodulatory therapy is used in refractory VAs. Despite transepidural anesthesia being a theoretically more effective method, it is limited by a concern for intrathecal bleeding risk. Renal nerve denervation, however, like other neuromodulation, requires a definite treatment endpoint, for which aorticorenal ganglion ablation could be the direction. However, the current evidence has shown how little we know about this delicate system and how to rebalance it. Additionally, most neuromodulatory interventions still require a reliable parameter for successful neuromodulation and specific stimulation settings. Nevertheless, individual variations, including age and comorbidities, make the refinement of neuromodulation difficult, and the relatively young age of this field limited the long-term follow-up studies. Although the field has a promising future for managing complex VAs, we are not there yet.

## Data Availability

No new data were created or analyzed in this study. Data sharing is not applicable to this article.

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
