# Peer review of "Sympathetic Modulation in Cardiac Arrhythmias: Where We Stand and Where We Go"

_jpm, 2023, doi:10.3390/jpm13050786_

Round 1

Reviewer 1 Report

The present paper is an interesting short review, regarding the sympathetic nervous system modulation in the context of cardiac arrhythmias. However, in order to consider it for publishing, the following should be taken into account:

1.      For section 3 please denote subsections 3.1, 3.2, etc.

2.      Section 3 (Assessing autonomic nervous system (ANS) function), subsection "Heart rate variability" is extremely brief and rather a short history section, than a review on the evaluation of heart rate variability parameters in the context of the sympathetic nervous system. The subsection must be redone.

3.      I think it would be interesting to briefly discuss some situational conditions associated with increased sympathetic tone and consequently increased risk of rhythm disorders (e.g., physical stress/exertion, emotional stress or sleep disorders).

4.      For section 7, subsection 7.1, I think a discussion is also necessary about ivabradine and its involvement in modulating the autonomic nervous system, which can automatically participate in changing the response in case of some arrhythmias, especially supraventricular ones. I think it is worth discussing, among others, the following: a. Scridon A. Long-Term Effects of Ivabradine on Cardiac Vagal Parasympathetic Function in Normal Rats. Front Pharmacol. 2021 Apr 8;12:596956. doi: 10.3389/fphar.2021.596956. b. Kawada T. Ivabradine preserves dynamic sympathetic control of heart rate despite inducing significant bradycardia in rats. J Physiol Sci. 2019 Mar;69(2):211-222. doi: 10.1007/s12576-018-0636-2.

Minor editing of English language required

Author Response

Dear reviewer

We have provided a point-by-point response to the reviewer's comments in the attached file.

Reviewer 2 Report

Thank you for the opportunity to review this manuscript by Chung et al, “Sympathetic modulation on cardiac arrhythmias: Where we stand and where we go.” This is a fairly comprehensive review describing the influence of the autonomic nervous system on cardiac rhythm disorders, and therapeutic options to influence this complex interaction. The manuscript does a nice job of explaining various complex mechanisms involved, particularly in the setting of cardiac disease. 

I have the following specific comments: 

1. In section 3, on Heart rate variability, the concept of “low-frequency and high-frequency" was discussed, but seems unclear what this is referring to, and would benefit from more explanation.  

2. Also in section 3 on Baroreflex sensitivity, the discussion of the ATRAMI study was also unclear and contains grammatical errors; see “...the standard deviation of the average of the normal sinus to normal sinus intervals …" 

3. The discussion of various therapeutic modalities is very useful and clinically appropriate. However, there seemed to be missing specific discussion of recently adopted more common clinical strategies; in the section on Carotid baroreceptor stimulation, a discussion of the recently approved CVRx/Barostim device should be included, including clinical data supporting this. In addition, “Cardioneuroablation” is now a more commonly performed procedure to ablate parasympathetic inputs into the heart via a percutaneous catheter based approach.  

4. The use of multiple abbreviations to describe the various therapeutic strategies is confusing. In addition, in the conclusion it appears the abbreviation BSG is used instead of SGB. I suggest not abbreviating these various interventions in order to improve reader comprehension.  

The manuscript is well written

Author Response

(The authors gave the same response as above.)

Round 2

Reviewer 1 Report

In my opinion, the changes made by the authors according to the suggestions are sufficient to suggest the acceptance of the paper for publication in its present form.

Only minor wording changes are needed.